# Late Evening Snack with Branched-Chain Amino Acids Supplementation Improves Survival in Patients with Cirrhosis

**DOI:** 10.3390/jcm9041013

**Published:** 2020-04-03

**Authors:** Tatsunori Hanai, Makoto Shiraki, Kenji Imai, Atsushi Suetsugu, Koji Takai, Masahito Shimizu

**Affiliations:** 1Department of Gastroenterology/Internal Medicine, Gifu University Graduate School of Medicine, 1-1 Yanagido, Gifu 501-1194, Japan; hanai@gifu-u.ac.jp (T.H.); ikenji@gifu-u.ac.jp (K.I.); asue@gifu-u.ac.jp (A.S.); koz@gifu-u.ac.jp (K.T.); shimim-gif@umin.ac.jp (M.S.); 2Department of Division for Regional Cancer Control, Gifu University Graduate School of Medicine, 1-1 Yanagido, Gifu 501-1194, Japan

**Keywords:** branched-chain amino acids, cirrhosis, late evening snack, liver failure, malnutrition, nutritional supplementation, survival

## Abstract

The clinical efficacy of a late evening snack (LES) is well documented in patients with cirrhosis, but its effect on survival remains unclear. This cohort study aimed to compare the overall survival between LES-treated patients and untreated patients. We conducted a retrospective cohort study to determine the effect of LES, which is defined as an oral intake of a branched-chain amino acids (BCAA)-enriched nutrient before bedtime, on survival in 523 patients with cirrhosis seen at a tertiary referral center in Japan from March 2004 to April 2019. The association between LES and all-cause mortality was evaluated using propensity score matching and inverse probability of treatment weighting analyses. The median age of the 523 participants was 66 years; 286 (55%) patients were men and 87 (17%) received LES therapy. Of the 231 propensity-matched patients, 20 (26%) LES-treated patients and 72 (47%) untreated patients died during a median follow-up of 2.0 years (interquartile range, 0.5–4.8). Propensity score matching analysis showed that the overall survival was significantly higher in LES-treated patients than in untreated patients (hazard ratio [HR], 0.57; 95% confidence interval [CI], 0.34–0.93). The survival benefit of LES therapy was most prominent in patients with Child–Pugh C cirrhosis (HR, 0.40; 95% CI, 0.20–0.81). Inverse probability of treatment weighting analysis also revealed that LES significantly improved the prognosis of patients with cirrhosis (HR, 0.57; 95% CI, 0.33–0.99). In this retrospective study of patients with cirrhosis, we found that nocturnal BCAA supplementation was associated with a significant reduction in the risk of death in patients with liver cirrhosis.

## 1. Introduction

Malnutrition is a frequent complication observed in patients with cirrhosis, occurring in 20%–50% patients depending on the degree of liver dysfunction [1]. This poor nutritional status or nutritional deficiencies is associated with the development, severity, and increased rates of complications including infections, hepatic encephalopathy (HE), ascites, and sarcopenia, all of which impair the prognosis of patients with cirrhosis [1,2,3,4,5]. Reduced dietary intake contributes to malnutrition in patients with cirrhosis [6]. Insufficient dietary intake increases the risk of morbidity and mortality, whereas sufficient dietary intake can improve clinical outcomes [7]. Patients with malnutrition frequently require nutritional intervention to improve their clinical outcomes.

Cirrhosis is a state of accelerated starvation which is characterized by the change of main energy sources from glucose to lipid use during the fasting state [8]. Due to the decrease of hepatic glycogen, patients with cirrhosis have increased rates of fat oxidation and gluconeogenesis, whereas such patients exhibit decreased rates of glucose oxidation and glycogenolysis, compared to normal individuals [8]. Since this metabolic profile is equivalent to that observed in healthy individuals after 2–3 days of starvation, taking snacks before bedtime or during nighttime hours is recommended for patients with cirrhosis to shorten the overnight fasting period [1]. Studies investigating the influence of meal patterns on energy metabolism have shown that a late evening snack (LES), which is defined as an oral intake of a branched-chain amino acids (BCAA)-enriched nutrient before bedtime, and more frequent oral intake improve the catabolic state of patients with cirrhosis [8]. A LES has been reported to improve not only the nutritional status but also clinical outcomes, including ascites, HE, and health-related quality of life in patients with cirrhosis [8,9,10]. Therefore, clinical practice guidelines recommend that patients with cirrhosis receive LES therapy to improve their clinical outcomes [1,7,11].

Although a LES is widely accepted to be an effective nutritional therapy for patients with cirrhosis with a high risk of malnutrition, to the best of our knowledge, no clinical trial has attempted to determine whether a LES improves the prognosis of patients with cirrhosis. LES is a clinically recommended and commonly accepted nutritional intervention for patients with cirrhosis; however, randomized controlled trials (RCTs) to investigate its survival benefit are apparently unethical and unfeasible. Therefore, we comprehensively analyzed our clinical data using propensity score (PS) matching and inverse probability of treatment weighting (IPTW), both of which can decrease bias in estimating treatment effects and reduce the likelihood of confounding when analyzing observational data [12]. We subsequently assessed the effect of a LES on the survival of patients with cirrhosis.

## 2. Materials and Methods

### 2.1. Study Design

We conducted a retrospective cohort study to evaluate the effect of a LES on survival in patients with cirrhosis. Considering the retrospective nature of the study, an opt-out approach was used to obtain informed consent from the patients, and personal information was strictly protected during data collection. The study protocol was reviewed and approved by the ethics committee of the Gifu University Graduate School of Medicine (approval number: 2019-0501). The study was performed in accordance with the Declaration of Helsinki and Good Clinical Practice guidelines.

### 2.2. Study Patients

A total of 523 patients who were admitted to Gifu University Hospital (Gifu, Japan) between March 2004 and April 2019 were potentially eligible for this study. The inclusion criteria were patients aged 20 years or older with liver cirrhosis of any etiology. The exclusion criteria included patients with hepatocellular carcinoma (HCC) and other active malignancies, liver transplantation and other organ transplantations, extrahepatic organ failure, amino acid metabolism disorders, oral intake difficulty, and malabsorption.

We categorized patients into a LES group (patients treated with LES) and a no LES group (patients not given LES therapy). In this report, we defined the term “LES” as daily use of a BCAA-enriched powder mix (Aminoleban EN powder mix; Otsuka Pharmaceutical Co., Ltd., Tokyo, Japan) before bedtime or during nighttime hours. Less frequent use or never-use was defined as “No LES.” The powder mix contained 213 kcal of total energy; 31.5 g of glucose; 13.5 g of proteins, including 6.1 g of BCAA (1.602 g of L-valine, 2.037 g of L-leucine, and 1.923 g of L-isoleucine); and 3.7 g of lipids, electrolytes, trace minerals, and vitamins. In our study, the decision to initiate LES supplementation depended completely on each physician. In general, according to the clinical practical guidelines for cirrhosis [11], we gave LES therapy to patients with cirrhosis who were unable to take sufficient dietary intake, and/or who suffered from complications including malnutrition, HE, and ascites. We collected information on the timing of LES supplementation from the electronic medical records of our hospital and found that most LES-treated patients took this nutrient between 09:00 p.m. and 11:00 p.m.

### 2.3. Clinical Characteristics

We collected information on the clinical characteristics and laboratory parameters at study entry. Variables included age, sex, body mass index (BMI), cirrhosis etiology, ascites, HE, serum levels of alanine aminotransferase, albumin, creatinine, sodium, total bilirubin, international normalized ratio (INR), platelet count, Child–Pugh score, and model for end-stage liver disease (MELD) score.

Of the 523 patients, a total of 463 patients were eligible to evaluate the cross-sectional area of abdominal skeletal muscles at the third lumbar vertebra, which accurately represents the whole-body skeletal muscle mass [4]. Sarcopenia was diagnosed as described in the literature [13].

Cirrhosis was diagnosed based on a combination of clinical characteristics including histological features, laboratory parameters, clinical features of portal hypertension, endoscopic findings of varices, and/or medical imaging features. Ascites were evaluated by medical imaging, including abdominal ultrasonography, computed tomography, and/or magnetic resonance imaging. HE was evaluated by each physician according to the West Haven criteria [14]. The degree of liver dysfunction was assessed by Child–Pugh and MELD scores [1]. Laboratory parameters were estimated using standard methods at the clinical laboratory of Gifu University Hospital.

### 2.4. Clinical Care and Follow-Up

All patients were treated according to the clinical practical guidelines for cirrhosis [11]. Patients were followed up every 1 to 3 months at our outpatient department. Clinical assessments included physical examinations, laboratory tests, and liver-related events, including gastrointestinal bleeding, ascites, spontaneous bacterial peritonitis, HE, HCC, and death. HCC was screened using imaging modalities such as abdominal ultrasonography, computed tomography, and/or magnetic resonance imaging. When HCC was diagnosed, all patients were treated, whenever possible, according to the practical guidelines for HCC [15].

### 2.5. Outcome

The primary outcome was death from all causes. The overall survival times were calculated based on the interval between entry and the last visit, date of death, or October 31, 2019, whichever occurred first. We used PS matching and IPTW methods to ensure comparability between the LES and No LES groups [12], and we subsequently assessed the significance of survival between the two groups.

### 2.6. Statistical Analysis

Baseline characteristics were shown as median with interquartile range (IQR) for continuous variables and number with percentages (%) for categorical variables. Differences between the study groups were analyzed using the Mann–Whitney *U* test for continuous variables and Pearson’s chi-square test for categorical variables. The overall survival was estimated using the Kaplan–Meier method and compared between groups using the log-rank test.

Since the lack of randomization and differences in the baseline characteristics between the two groups may have influenced the outcomes, we used PS matching and IPTW analyses to estimate the clinical outcomes [12,16]. The PS was calculated using a logistic regression model with variables, including age, sex, BMI, cirrhosis etiology, ascites, HE, alanine aminotransferase, albumin, creatinine, sodium, total bilirubin, INR, platelet count, Child–Pugh score and class, and MELD score. This model yielded a c-index of 0.782 (95% confidence interval [CI], 0.736–0.828). The PS was assigned to each individual patient and a 1:2 match between the LES and No LES groups was performed by nearest-neighbor matching without replacement, with a caliper of width equal to 0.2 of the standard deviation of the logit of the PS [17]. IPTW using the PS allowed us to obtain unbiased estimates of the average treatment effects and include all study participants in the analysis [16]. Statistical weights were estimated as 1/PS for LES-treated patients and 1/(1-PS) for untreated patients [16]. We used standardized mean differences to quantitatively compare the balance in baseline variables between the LES and No LES groups, and although there is no universally accepted threshold, a value of <0.1 is reported to be well balanced [12].

We used Cox proportional hazards regression to evaluate the association between LES and all-cause mortality, and the results were expressed as the hazard ratio (HR) with 95% CI. All tests were two-sided, and the significance threshold was set at 0.05. All statistical analyses were performed using R version 3.6.1 (The R Foundation for Statistical Computing, Vienna, Austria)

## 3. Results

### 3.1. Baseline Characteristics of Patients

Among the 523 patients, 286 (55%) were men, with a median age of 66 years (IQR, 57–74) and a median BMI of 22.7 kg/m^2^ (IQR, 21.0–25.9). Of the study patients, 87 (17%) were treated with LES and 436 (83%) did not receive LES therapy. The baseline characteristics of the LES-treated and untreated patients are detailed in Table 1.

No significant difference was noted in the age, sex, and BMI between the two groups, but the prevalence of ascites and HE was significantly higher in the LES-treated patients than the untreated patients. Compared to untreated patients, LES-treated patients had significantly more advanced liver disease in terms of laboratory variables (albumin, sodium, INR, and platelet count) and liver function (Child–Pugh score and class and MELD score).

PS matching analysis identified 77 LES-treated patients and 154 PS-matched patients; this model showed no significant difference in patients’ characteristics between the two groups (Table 2 and Appendix A).

Of the 523 patients, a total of 463 patients were eligible to evaluate skeletal muscle mass and sarcopenia. The results found that LES-treated patients had a significantly less muscle mass than untreated patients (*p* = 0.039). After PS-matching analysis, no significant difference was noted in muscle mass (*p* = 0.941) and sarcopenia (*p* = 0.885) between the two groups.

### 3.2. Impact of a LES on The Survival of Patients With Cirrhosis

During a median follow-up of 2.4 years (IQR, 0.7–5.5), 21 (24%) patients in the LES group and 142 (33%) in the No LES group died. No patient underwent liver transplantation during the follow-up period. Before PS matching analysis, no significant difference in overall survival rate was noted between the LES and No LES groups (median, 9.7 vs. 7.8 years; P = 0.38; Figure 1a), with no reduction in the risk of mortality in the LES group (HR, 0.81; 95% CI, 0.51–1.29).

After PS matching analysis, 20 (26%) patients in the LES group and 72 (47%) in the No LES group died. The causes of death of the patients in both groups before and after PS matching are summarized in Appendix A. The 1-, 3-, 5-, and 10-year probabilities of survival in the LES group were 89%, 75%, 68%, and 42%, respectively, and those in the No LES group were 79%, 60%, 48%, and 34%, respectively. The overall survival rate was significantly higher in the LES group than in the No LES group (median, 5.8 vs. 4.5 years; *p* = 0.025; Figure 1b), resulting in a 43% reduction in the risk of mortality in the LES group (HR, 0.57; 95% CI, 0.34–0.93). Additionally, subgroup analysis showed that a LES improved prognosis, especially in patients with Child–Pugh C cirrhosis (HR, 0.40; 95% CI, 0.20–0.81; *p* = 0.011), whereas no survival benefit of a LES was found in patients with Child–Pugh A (HR, 0.43; 95% CI, 0.09–1.95; *p* = 0.27) and Child–Pugh B cirrhosis (HR, 0.69; 95% CI, 0.30–1.57; *p* = 0.37; Figure 2a–c).

In IPTW analysis, the baseline characteristics were similar between the LES and No LES groups (Appendix A). Cox regression analysis with IPTW also showed that a LES was significantly associated with a reduction in the risk of mortality (HR, 0.57; 95% CI, 0.33–0.99; *p* = 0.046).

Since many studies have shown that diabetes, which is one of the most common complications of cirrhosis [18], is associated with increased risk of mortality [19], we analyzed the impact of diabetes on mortality in patients with cirrhosis. Of the 231 PS-matched patients, diabetes was presented in 58 patients (25%): 12 were in the LES group and 46 were in the No LES group (*p* = 0.018). The results found that there was no association between diabetes and mortality (HR, 0.93; 95% CI, 0.57–1.48; *p* = 0.78), and the subgroup analysis also confirmed the same results both in the LES (HR, 0.65; 95% CI, 0.10–2.30; *p* = 0.55) and the No LES groups (HR, 0.88; 95% CI, 0.52–1.45; *p* = 0.63).

### 3.3. Effect of Daytime Versus Bedtime BCAA Supplementation on The Survival

To determine which timing of BCAA supplementation can lead to better survival in patients with cirrhosis, we compared the survival rates between patients treated with LES and those given daytime BCAA supplementation. Among the 523 patients, 157 (30%) received daytime BCAA supplementation and 87 (17%) were given LES supplementation. The results showed that the overall survival rate was significantly higher in the LES group than in the daytime group (median, 9.7 vs. 3.3 years; *p* < 0.001; Figure 3a), with the HR of 0.40 (95% CI, 0.24–0.63). Of the 231 PS-matched patients, 100 (43%) received daytime BCAA supplementation and 77 (33%) were given LES supplementation. The overall survival rate was significantly higher in the LES group than in the daytime group (median, 5.8 vs. 2.5 years; *p* < 0.001; Figure 3b). The results showed that LES supplementation significantly improved prognosis, compared to daytime BCAA supplementation (HR, 0.42; 95% CI, 0.24–0.68; *p* < 0.001; Figure 3b).

### 3.4. LES and Changes in Child–Pugh Score

We further analyzed clinical data of 146 propensity-matched patients (49 were in the LES group and 97 were in the No LES group) who were followed up for more than one-year, and the changes in Child–Pugh score at baseline, 6, and 12 months were evaluated to clarify the effect of LES on the severity of liver dysfunction. The mean Child–Pugh scores at baseline, 6, and 12 months were 8.1, 6.6, and 6.1, respectively, in the LES group, whereas those were 7.5, 7.5, and 7.9, respectively, in the No LES group. Repeated measures ANOVA revealed that although the Child–Pugh score was significantly decreased in the LES group during the follow-up period (*p* < 0.001), this score was statistically increased in the No LES group (*p* < 0.001; Figure 4).

## 4. Discussion

This study is the first to determine the association between LES therapy and improved survival in patients with cirrhosis. A LES has been shown to improve the nutritional status, liver function reserves, and sarcopenia in patients with cirrhosis [8,9,10], each of which raises the potential to improve the survival of these patients; however, little is known about the survival benefit of LES thus far. The novelty of our study is that LES-treated patients demonstrated a significantly higher overall survival than PS-matched patients who did not receive LES therapy. More importantly, this survival benefit was more significant in patients with Child–Pugh C cirrhosis than in those with Child–Pugh A or B cirrhosis. To the best of our knowledge, our findings provide the first evidence that LES can reduce the risk of mortality in this patient population. These findings underscore the importance of providing LES therapy to patients with cirrhosis with a high risk of malnutrition.

A LES is now clinically accepted as an effective nutritional intervention for patients with cirrhosis [1,7,11]; however, the majority of studies examining the efficacy of LES are likely to be biased by a relatively small sample size, short-observation period, and lack of a randomization or control group [8]. Long-term RCTs that directly compare LES-treated patients to untreated patients would be ideal, but are difficult to perform since withholding a LES for many years is both unfeasible and unethical. Therefore, we attempted to reduce bias using PS matching and IPTW analyses, both of which allowed us to conduct a hypothetical interventional study and subsequently compare clinical outcomes between the two groups. For these reasons, despite the lack of actual randomization, our findings present solid evidence that a LES is associated with improved prognosis in patients with cirrhosis.

The survival benefit of a LES in this study can be inferred from its timing, composition, dose, and duration. First, many studies have evaluated which time of nutritional supplementation can improve clinical outcomes of patients with LC effectively [8]. They have demonstrated that a LES, compared to daytime supplementation, can contribute to an improvement in nutritional status, despite an equal total energy intake in both groups [20,21,22]. Second, the LES composition may affect the outcome of patients with LC. Several studies have investigated the efficacy of a LES using various compositions, including liquid nutritional supplements, high-carbohydrate foods (e.g., bread and jam, rice ball, or oral glucose), and BCAA-enriched supplements [8,20]. Of these LES compositions, BCAA-enriched nutrients are considered more effective in patients with cirrhosis since BCAA serves as a substrate not only for energy generation but also for both protein synthesis and ammonia detoxification in skeletal muscle [1,2,3,23]. Large-scale clinical trials on BCAA supplementation have proven that it improves not only nutritional status but also the prognosis of patients with cirrhosis [24,25,26]. Additionally, several RCTs have shown that a LES of a BCAA-enriched nutrient can improve the nutritional status and quality of life, compared to a LES of ordinary food [20]. Thus, the use of BCAA is likely to be biologically plausible and an optimal composition for LES therapy. Third, most studies on LESs are conducted using a LES with a total calorie content of 150–350 kcal [8]. Some studies have used a LES with a high calorie content of 710 kcal and shown an increase in body protein stores [8,21]; however, further studies are required to establish the optimal calorie content of a LES.

Fourth, there is limited information on the relationship between LES duration and clinical outcomes in patients with cirrhosis. Plank LD and colleagues have performed a 12-month randomized controlled trial to determine whether nocturnal nutritional supplementation can improve body protein stores in patients with cirrhosis [21]. A total of 103 patients were randomly assigned to daytime or bedtime nutritional supplementation group, and total body protein at baseline, 3, 6, and 12 months were evaluated. Despite the fact that the amount of total body protein at baseline was equivalent in each group, the results have shown that total body protein was significantly increased in the bedtime supplementation group over 12 months, whereas no significant increase in total body protein were noted in the daytime group. Although they showed significant improvement in muscle mass and nutritional status in patients with cirrhosis, they demonstrated no significant benefits in survival [21], compared to our study. Poor adherence with the prescribed nutritional regimen in their study may explain the poor survival benefit of LES therapy. In addition, this discrepancy may be explained by the larger number of patients, longer follow-up period, more advanced liver disease, and the different primary outcome and methodology in our study. LES-treated patients in our study were provided LES therapy with the appropriate timing, BCAA-enriched nutrient, adequate calorie content, and long-term duration, and all the aforementioned factors likely underlie the higher overall survival in LES-treated patients.

The use of BCAA as a nutritional supplementation may contribute to the positive effects of LES on survival in patients with cirrhosis. We, therefore, compared the survival rates between patients treated with LES and those given daytime BCAA supplementation, and found that LES supplementation, compared to daytime supplementation, clearly improves prognosis in patients with cirrhosis. These results suggest that nocturnal BCAA supplementation, rather than daytime BCAA supplementation, may contribute to the improvement of survival, although the retrospective nature of this study limits the conclusion of the survival benefit of LES therapy. Further prospective studies are therefore required to validate the effect of LES therapy on survival in patients with cirrhosis.

This study has several limitations. First, although PS matching and IPTW analyses can compensate for the lack of randomization and reduce bias and measured confounders [12], residual bias and unmeasured confounders may still exist. Furthermore, the retrospective nature of our study limits the assessment of variables including dietary intake, daily physical activity or exercise, and weight change, all of which may affect outcomes in patients with chronic liver disease [3,6,27]. It, therefore, follows from this point that a RCT is optimal to determine an effect of intervention on outcomes [12]. Second, we lack definite information on adherence, which also affects clinical outcomes [28]. A multicenter prospective cohort study on BCAA supplementation revealed that poor adherence (approximately half or less) was associated with a risk of progression of liver failure, whereas good adherence (nearly all) improved the event-free survival in patients with cirrhosis [29]. Regarding adherence to LES therapy, most clinical trials reported from Japan have shown good adherence, ranging from 81% to 100% [8,20,22,29]. Although the reason for this high adherence remains unclear, the use of a BCAA-enriched nutrient as a LES supplementation may contribute to good adherence since the BCAA-enriched nutrient that is clinically approved in Japan has improved taste and palatability. We, therefore, consider that the influence of LES adherence on the outcomes of our study is relatively small. The lack of data concerning compliance is a critical limitation of this study. Due to the retrospective nature of this study, we had no data on compliance. Therefore, a prospective study should be performed to validate the findings of the present study. Finally, our study is based on a single-center cohort of patients, which has a potential limitation to generalize our findings to other populations and regions. Therefore, further multicenter and prospective cohort studies are warranted to clarify the causal relationship between LES and survival.

We believe that these limitations are outweighed by the strengths of our study, including the large cohort of patients, longer follow-up period, and the use of two robust statistical analyses which allows us to manage the concern of lack of randomization and to demonstrate our hypothesis. Our study provides clinically valid information in that our findings shed new light on the relationship between LES therapy and improved survival, and we consider that LES has the potential to hold great promise as an effective nutritional intervention for patients with cirrhosis with a high risk of malnutrition.

## 5. Conclusions

In conclusion, nocturnal BCAA supplementation can improve the prognosis of patients with cirrhosis, especially those with advanced cirrhosis. Overall, our findings suggest that a LES is most likely clinically beneficial for patients with cirrhosis.

## Figures and Tables

**Figure 1 jcm-09-01013-f001:**
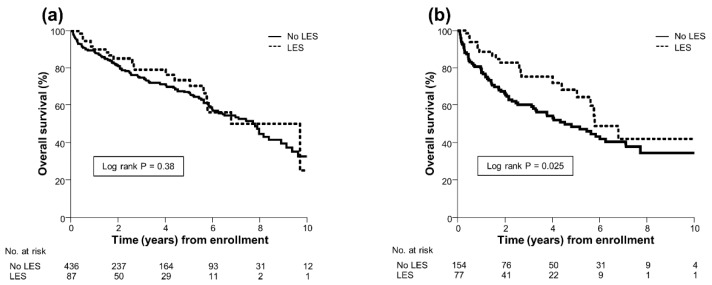
(**a**) Overall survival of 87 LES-treated patients and 436 untreated patients. (**b**) Overall survival of 77 LES-treated patients and 154 untreated patients after propensity score matching analysis. Overall survival was estimated using the Kaplan–Meier method and compared between groups using the log-rank test. LES, late evening snack.

**Figure 2 jcm-09-01013-f002:**
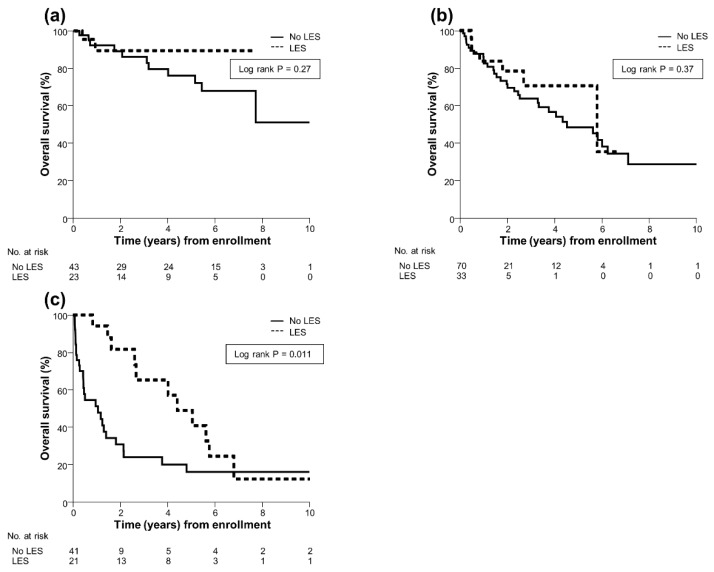
(**a**) Overall survival of 23 LES-treated patients and 43 untreated patients with Child–Pugh A cirrhosis. (**b**) Overall survival of 33 LES-treated patients and 70 untreated patients with Child–Pugh B cirrhosis. (**c**) Overall survival of 21 LES-treated patients and 41 untreated patients with Child–Pugh C cirrhosis. Overall survival was estimated using the Kaplan–Meier method and compared between groups using the log-rank test. LES, late evening snack.

**Figure 3 jcm-09-01013-f003:**
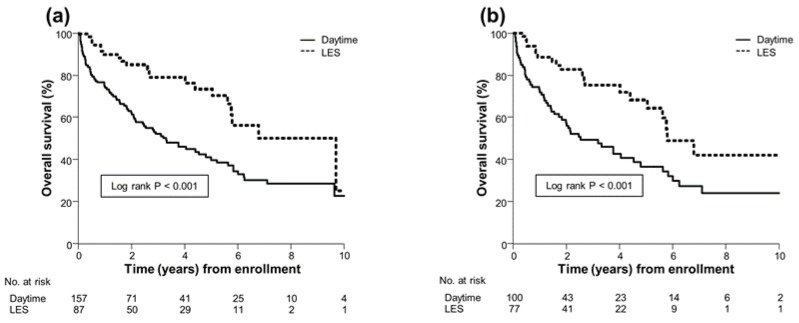
(**a**) Overall survival both of 87 patients treated with LES and 157 patients given daytime BCAA supplementation. (**b**) Overall survival both of 77 patients treated with LES and 100 patients given daytime BCAA supplementation after propensity score matching analysis. Overall survival was estimated using the Kaplan–Meier method and compared between groups using the log-rank test. BCAA, branched-chain amino acids; LES, late evening snack.

**Figure 4 jcm-09-01013-f004:**
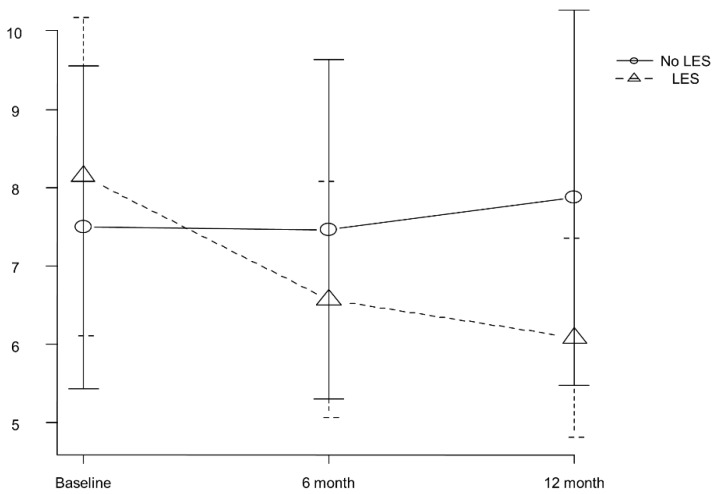
Changes in Child–Pugh score at baseline, 6, and 12 months in the LES and No LES groups. Data was analyzed by repeated measures ANOVA. LES, late evening snack.

**Table 1 jcm-09-01013-t001:** Clinical characteristics of 87 LES-treated patients and 436 untreated patients.

	No LES †	LES †	*p*-Value *	SMD ‡
Characteristic	(n = 436)	(n = 87)		
Age (years)	66.0 (55.0–74.0)	69.0 (59.0–74.0)	0.192	0.204
Men	237 (54.4)	49 (56.3)	0.814	0.040
BMI (kg/m^2^)	22.9 (21.0–25.3)	22.5 (21.1–24.5)	0.224	0.157
Etiology of cirrhosis				
HBV	22 (5.0)	11 (12.6)	<0.001	0.534
HCV	200 (45.9)	31 (35.6)		
ALD	75 (17.2)	29 (33.3)		
Others	139 (31.9)	16 (18.4)		
Ascites	150 (34.4)	59 (67.8)	0.001	0.709
Hepatic encephalopathy	27 (6.2)	12 (13.8)	0.023	0.255
ALT (IU/L)	35 (21–50)	23 (16–36)	<0.001	0.485
Albumin (g/dL)	3.5 (3.0–4.0)	3.0 (2.6–3.4)	<0.001	0.767
Creatinine (mg/dL)	0.72 (0.57–0.88)	0.67 (0.56–0.91)	0.707	0.011
Sodium (mEq/L)	139 (137–141)	138 (136–140)	0.002	0.259
Total bilirubin (mg/dL)	1.00 (0.80–1.60)	1.10 (0.90–2.00)	0.155	0.083
Platelet (10⁹/L)	103 (71–161)	83 (54–118)	0.001	0.138
INR	1.08 (1.00–1.21)	1.16 (1.04–1.30)	0.002	0.048
MELD score	8 (7–11)	10 (7–12)	0.006	0.207
Child–Pugh score	6 (5–8)	8 (6–10)	< 0.001	0.643
Child–Pugh class				
A	267 (61.2)	23 (26.4)	<0.001	0.750
B	107 (24.5)	39 (44.8)		
C	62 (14.2)	25 (28.7)		
Skeletal muscle index (cm^2^/m^2^) **	44.1 (37.8–51.2)	41.9 (36.5–47.9)	0.039	0.299
Sarcopenia **	200 of 380 (52.6)	50 of 83 (60.2)	0.226	0.154

Values are presented as numbers (percentages) or medians (interquartile ranges). ALD, alcohol-related liver disease; ALT, alanine aminotransferase; BMI, body mass index; HBV, hepatitis B virus; HCV, hepatitis C virus; INR, international normalized ratio; LES, late evening snack; MELD, model for end-stage liver disease; SMD, standardized mean differences. † LES was defined as daily use of a branched-chain amino acid-enriched powder mix before bedtime. Less frequent use or never-use was defined as No LES. * The chi-square test for categorical variables or Mann–Whitney *U* test for continuous variables were used to compare the clinical characteristics between the two groups. ** 463 eligible patients. ‡ SMD was used to compare the balance in baseline variables between the two groups.

**Table 2 jcm-09-01013-t002:** Clinical characteristics of 77 LES-treated patients and 154 untreated patients after propensity score matching analysis.

	No LES †	LES †	*p*-Value *	SMD ‡
Characteristic	(n = 154)	(n = 77)		
Age (years)	68.5 (60.0–75.0)	68.0 (59.0–74.0)	0.646	0.014
Men	83 (53.9)	45 (58.4)	0.575	0.092
BMI (kg/m^2^)	22.1 (20.5–24.7)	22.3 (21.0–24.5)	0.752	0.070
Etiology of cirrhosis				
HBV	12 (7.8)	6 (7.8)	1.000	<0.001
HCV	60 (39.0)	30 (39.0)		
ALD	50 (32.5)	25 (32.5)		
Others	32 (20.8)	16 (20.8)		
Ascites	97 (63.0)	50 (64.9)	0.885	0.041
Hepatic encephalopathy	19 (12.3)	12 (15.6)	0.541	0.094
ALT (IU/L)	26 (17–38)	23 (16–39)	0.304	0.026
Albumin (g/dL)	3.0 (2.5–3.4)	3.0 (2.7–3.4)	0.457	0.080
Creatinine (mg/dL)	0.75 (0.61–0.94)	0.68 (0.58–0.92)	0.399	0.017
Sodium (mEq/L)	138 (136–140)	138 (136–140)	0.625	0.010
Total bilirubin (mg/dL)	1.35 (0.80–2.10)	1.00 (0.80–1.90)	0.362	0.075
Platelet (10⁹/L)	92 (68–124)	87 (55–119)	0.206	0.044
INR	1.15 (1.06–1.30)	1.14 (1.04–1.30)	0.492	0.055
MELD score	10 (8–13)	9 (7–12)	0.348	0.085
Child–Pugh score	8 (6–10)	8 (6–10)	0.950	0.037
Child–Pugh class				
A	43 (27.9)	23 (29.9)	0.936	0.055
B	70 (45.5)	33 (42.9)		
C	41 (26.6)	21 (27.3)		
Skeletal muscle index (cm^2^/m^2^) **	42.4 (36.2–47.9)	41.9 (36.3–48.4)	0.941	0.029
Sarcopenia**	87 of 149 (58.4)	44 of 73 (60.2)	0.885	0.074

Values are presented as numbers (percentages) or medians (interquartile ranges). ALD, alcohol-related liver disease; ALT, alanine aminotransferase; BMI, body mass index; HBV, hepatitis B virus; HCV, hepatitis C virus; INR, international normalized ratio; LES, late evening snack; MELD, model for end-stage liver disease; SMD, standardized mean differences. † LES was defined as daily use of a branched-chain amino acid-enriched powder mix before bedtime. Less frequent use or never-use was defined as No LES. * The chi-square test for categorical variables or Mann–Whitney *U* test for continuous variables were used to compare the clinical characteristics between the two groups. ** 222 eligible patients. ‡ SMD was used to compare the balance in baseline variables between the two groups.

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
