# Peer review of "Late Evening Snack with Branched-Chain Amino Acids Supplementation Improves Survival in Patients with Cirrhosis"

_jcm, 2020, doi:10.3390/jcm9041013_

Round 1

Reviewer 1 Report

GENERAL COMMENT

Nutrition has not been a primary focus in end-stage liver disease despite its importance in the development, severity and increased rates of complications and mortality in cirrhosis owing to nutritional deficiencies (Clin Liver Dis. 2014 Feb;18(1):179-90). On this background, it is logical to postulate that interventions aimed at correcting such deficiencies will result into improved survival in hepatic cirrhosis. Confirming this, doctor Hanai and Colleagues determined the effect of LES on survival in 523 patients with cirrhosis seen at a tertiary referral center in Japan. Data have shown that late evening snacks significantly improved the prognosis of patients with cirrhosis, particularly in patients with Child–Pugh C cirrhosis owing to various etiologies. This is an important though retrospective study. The mechanistic explanations of findings is poorly discussed and so is the rationale of the study. The role of diabetes should be better illustrated.

SPECIFIC COMMENT

The biological rationale of the study must be better developed. For example the following statement should be expanded by providing a synthetic account of the biochemical bases underlying this phenomenon: "Because cirrhosis is a state of accelerated starvation, taking snacks before bedtime or during nighttime hours is recommended for patients with cirrhosis to shorten the overnight fasting period".

Along the same line, I would suggest including into the statistical model also the variable "diabetes". Diabetes has a strong association with cirrhosis, which may be considered to be a diabetogenic condition. (Diabetes Res. 1988;7:185-8). A study conducted in Singapore found that Diabetes was a risk factor for mortality among those with cirrhosis (Liver Int. 2017;37:251-258). Therefore, it is important to understand which the role of diabetes was in the association of late evening snacks with improved survival.

Author Response

Responses to Reviewer 1

Thank you very much for reviewing our manuscript and offering valuable advice. We appreciate your comments, which have helped us to improve our manuscript. Please find below detailed responses to the reviewer’s comments.

  1. Nutrition has not been a primary focus in end-stage liver disease despite its importance in the development, severity and increased rates of complications and mortality in cirrhosis owing to nutritional deficiencies (Clin Liver Dis. 2014 Feb;18(1):179-90). On this background, it is logical to postulate that interventions aimed at correcting such deficiencies will result into improved survival in hepatic cirrhosis. Confirming this, doctor Hanai and Colleagues determined the effect of LES on survival in 523 patients with cirrhosis seen at a tertiary referral center in Japan. Data have shown that late evening snacks significantly improved the prognosis of patients with cirrhosis, particularly in patients with Child–Pugh C cirrhosis owing to various etiologies. This is an important though retrospective study. The mechanistic explanations of findings is poorly discussed and so is the rationale of the study. The role of diabetes should be better illustrated.

Thank you for the useful comments. We have revised the Introduction section and added new reference (lines 36 to 39; new Ref. 5).

  1. The biological rationale of the study must be better developed. For example the following statement should be expanded by providing a synthetic account of the biochemical bases underlying this phenomenon: "Because cirrhosis is a state of accelerated starvation, taking snacks before bedtime or during nighttime hours is recommended for patients with cirrhosis to shorten the overnight fasting period".

We agree that this study needs more information regarding the mechanism of the state of accelerated starvation in patients with cirrhosis. We have revised the manuscript based on the reviewer’s comments. The revised version is as follows: Cirrhosis is a state of accelerated starvation which is characterized by the change of main energy sources from glucose to lipid use during the fasting state. Due to the decrease of hepatic glycogen, patients with cirrhosis have increased rates of fat oxidation and gluconeogenesis, whereas such patients exhibit decreased rates of glucose oxidation and glycogenolysis, compared to normal individuals. Since this metabolic profile is equivalent to that observed in healthy individuals after 2–3 days of starvation, taking snacks before bedtime or during nighttime hours is recommended for patients with cirrhosis to shorten the overnight fasting period (lines 43 to 48).

  1. Along the same line, I would suggest including into the statistical model also the variable "diabetes". Diabetes has a strong association with cirrhosis, which may be considered to be a diabetogenic condition. (Diabetes Res. 1988;7:185-8). A study conducted in Singapore found that Diabetes was a risk factor for mortality among those with cirrhosis (Liver Int. 2017;37:251-258). Therefore, it is important to understand which the role of diabetes was in the association of late evening snacks with improved survival.

Since many studies have shown that diabetes, which is one of the most common complications of cirrhosis, is associated with increased risk of mortality, we analyzed the impact of diabetes on mortality in patients with cirrhosis. Of the 231 PS-matched patients, diabetes was presented in 58 patients (25%): 12 were in the LES group and 46 were in the No LES group (P = 0.018). The results found that there was no association between diabetes and mortality (HR, 0.93; 95% CI, 0.57–1.48; P = 0.78), and the subgroup analysis also confirmed the same results both in the LES (HR, 0.65; 95% CI, 0.10–2.30; P = 0.55) and the No LES groups (HR, 0.88; 95% CI, 0.52–1.45; P = 0.63) (lines 214 to 220; new Refs. 18 and 19). We thank your valuable comment.

In closing, let me thank you once again for your comments which have helped us to improve the quality of our paper.

Reviewer 2 Report

A well written paper with some interesting statistical methods which were well described

The late evening snack appears to be a branched chain amino acid supplement and this should be reflected in the title

The propensity score matching analysis demonstrates the positive effects of LES on survival and is an important point for clinicians

However, the effect is more than likely due to the BCAA supplementation and this must be more clearly stated in the title the abstract and the conclusion.

Author Response

Responses to Reviewer 2

Thank you very much for reviewing our manuscript and offering valuable advice. We appreciate your comments, which have helped us to improve our manuscript. Please find below detailed responses to the reviewer’s comments.

  1. A well written paper with some interesting statistical methods which were well described. The late evening snack appears to be a branched chain amino acid supplement and this should be reflected in the title.

Based on the reviewer’s comment, the title has been revised as follows: Late Evening Snack with Branched-Chain Amino Acids Supplementation Improves Survival in Patients with Cirrhosis.

  1. The propensity score matching analysis demonstrates the positive effects of LES on survival and is an important point for clinicians. However, the effect is more than likely due to the BCAA supplementation and this must be more clearly stated in the title the abstract and the conclusion.

We really understand that the use of BCAA as a nutritional supplementation may contribute to the positive effects of LES on survival in patients with cirrhosis. To determine which timing of BCAA supplementation can lead to better survival in patients with cirrhosis, we compared the survival rates between patients treated with LES and those given daytime BCAA supplementation. Among the 523 patients, 157 (30%) received daytime BCAA supplementation and 87 (17%) were given LES supplementation. The results showed that the overall survival rate was significantly higher in the LES group than in the daytime group (median, 9.7 vs. 3.3 years; P < 0.001; Figure 3a), with the HR of 0.40 (95% CI, 0.24–0.63). Of the 231 PS-matched patients, 100 (43%) received daytime BCAA supplementation and 77 (33%) were given LES supplementation. The overall survival rate was significantly higher in the LES group than in the daytime group (median, 5.8 vs. 2.5 years; P < 0.001; Figure 3b). The results showed that LES supplementation significantly improve prognosis, compared to daytime BCAA supplementation (HR, 0.42; 95% CI, 0.24-0.68; P < 0.001; Figure 3b). These results suggest that nocturnal BCAA supplementation, rather than daytime BCAA supplementation, may contribute to the improvement of survival, although the retrospective nature of this study limits the conclusion of the survival benefit of LES therapy. Further prospective studies are therefore required to validate the effect of LES therapy on survival in patients with cirrhosis. To state this point more clearly, we have revised the manuscript, accordingly (lines 2 to 3, 29, 221 to 232, 306 to 313; and new Figure 3). Thank you for your good suggestion.

In closing, let me thank you once again for your comments which have helped us to improve the quality of our paper.

Reviewer 3 Report

In the context of the scarcity of data and firm recommendations with regards to nutritional interventions in advanced liver disease, articles like the one submitted by Hanai T et al. are a welcomed sight, aiming to advance knowledge in the field further. However, the current form does have some significant caveats, which might lead to misinterpretation or confusion. I would like to start by highlighting some issues regarding study design. The current study aims to evaluate the relationship between nutritional intervention and all-cause mortality. While certainly interesting judging by the result, I find a lack of proper data continuity which can adequately explain the causality. First, there is a paucity of data at baseline assessing the nutritional status of the included subjects, the BMI being the only available variable (which can easily be misleading in advanced liver disease due to fluid retention). No other metrics or follow-up data were made available, other than the definite outcome. Furthermore, there is no data regarding the factors which prompted the decision to initiate the nutritional intervention. While it is obvious judging by the pre-PS matching that patients with the more advanced disease received LES, it would have been valuable to know what were the reasons for choosing which patients to treat (among the more advanced disease group). Did the patients from the PS match group receive other nutritional support, such as daytime snack/supplementation or parenteral supplementation? Was cost or patient preference a factor in the decision-making process? How was compliance accounted for if there was no other available data apart from in-hospital records (which is by definition short term)? Since the patients were followed-up every 1 to 3 months, it would add clarity if follow-up data were made available, including decompensation patterns and/or at least MELD/Child dynamics. Addressing the aforementioned caveats should add strength to the causal relationship if indeed there is one. Finally, the design and the results of the study could conclude that nutritional supplements, rather than LES (mainly because of lack of compliance verification), may improve survival. Furthermore, I would add some other miscellaneous/nuance observations regarding the paper: 1. The abstract includes the pre-PS mortality data (rows 19-20); post-PS data would certainly be a better fit. 2. The introduction should include a brief definition of LES (no more than a sentence, but expanding beyond de acronym). 3. Row 89-90 states that patients were classified as having ascites even though no ascites were detected if they were on diuretic treatment. I find this approach rather peculiar. In light of the current multistate/competing risk approach to advanced liver disease, I am not sure that it adds any clarity. 4. I would suggest expanding the comparative discussion with regards to the study published by Plank L et al. (Hepatology 2008, ref. 17 in the text) since this was a 12-month randomized trial that included more advanced nutritional metrics.

Author Response

Responses to Reviewer 3

Thank you very much for reviewing our manuscript and offering valuable advice. We appreciate your comments, which have helped us to improve our manuscript. Please find below detailed responses to the reviewer’s comments.

  1. The current study aims to evaluate the relationship between nutritional intervention and all-cause mortality. While certainly interesting judging by the result, I find a lack of proper data continuity which can adequately explain the causality. First, there is a paucity of data at baseline assessing the nutritional status of the included subjects, the BMI being the only available variable (which can easily be misleading in advanced liver disease due to fluid retention).

Concerning the comments on the lack of data regarding the nutritional status of the enrolled patients, we retrospectively assessed the skeletal muscle mass or sarcopenia that represents the nutritional status of patients with cirrhosis. Of the 523 patients, a total of 463 patients were eligible to evaluate the cross-sectional area of abdominal skeletal muscles at the third lumbar vertebra, which accurately represents the whole-body skeletal muscle mass. Sarcopenia was diagnosed as described in the literature (new Ref. 13). The results found that LES-treated patients had a significantly less muscle mass than untreated patients (P = 0.039). After PS-matching analysis, no significant difference was noted in muscle mass (P = 0.941) and sarcopenia (P = 0.885) between the two groups. We hope this answers the points raised by the reviewer. We have revised the Tables, Method, and Result sections (lines 99 to 101, 170 to 173: and revised Table 1 and 2).

  1. No other metrics or follow-up data were made available, other than the definite outcome.
  2. Since the patients were followed-up every 1 to 3 months, it would add clarity if follow-up data were made available, including decompensation patterns and/or at least MELD/Child dynamics. Addressing the aforementioned caveats should add strength to the causal relationship if indeed there is one.

We agree that follow-up data will support the strength of our study. Based on this suggestion, we further analyzed clinical data of 146 propensity-matched patients (49 were in the LES group and 97 were in the No LES group) who were followed up for more than one-year, and the changes in Child-Pugh score at baseline, 6, and 12 months were evaluated to clarify the effect of LES on the severity of liver dysfunction. The mean Child-Pugh scores at baseline, 6, and 12 months were 8.1, 6.6, and 6.1, respectively, in the LES group, whereas those were 7.5, 7.5, and 7.9, respectively, in the No LES group. Repeated measures ANOVA revealed that although the Child-Pugh score was significantly decreased in the LES group during the follow-up period (P < 0.001), this score was statistically increased in the No LES group (P < 0.001). These results were described in the Results section (lines 238 to 246 and new Figure 4).

  1. Furthermore, there is no data regarding the factors which prompted the decision to initiate the nutritional intervention. While it is obvious judging by the pre-PS matching that patients with the more advanced disease received LES, it would have been valuable to know what were the reasons for choosing which patients to treat (among the more advanced disease group).

We agree that there is a need to clarify the decision-making process. In our study, the decision to initiate LES supplementation depended completely on each physician. In general, according to the clinical practical guidelines for cirrhosis, we gave LES therapy to patients with cirrhosis who were unable to take sufficient dietary intake, and/or who suffered from complications including malnutrition, HE, and ascites (lines 87 to 90). We thank your valuable comment that improves the quality of our manuscript.

  1. Did the patients from the PS match group receive other nutritional support, such as daytime snack/supplementation or parenteral supplementation?

Of the 231 PS-matched patients, 100 (43%) received daytime BCAA supplementation and 77 (33%) were given LES supplementation. We have added this information in the revised manuscript (lines 227 to 228).

  1. Was cost or patient preference a factor in the decision-making process?

BCAA-enriched nutrient is widely available by health insurance in Japan. In addition, when patients take this nutrient, they can select flavors including yogurt, pineapple, apple, coffee, plum, green tea, and fruit. We, therefore, consider that the influence of cost and patient preference on the decision-making process was minimal.

  1. How was compliance accounted for if there was no other available data apart from in-hospital records (which is by definition short term)?

We really understand that the lack of data concerning compliance is critical limitation of this study. Unfortunately, due to the retrospective nature of this study, we had no data on compliance. Therefore, a prospective study should be performed to validate the findings of the present study (lines 329 to 331). We deeply appreciate your important suggestion.

  1. The design and the results of the study could conclude that nutritional supplements, rather than LES (mainly because of lack of compliance verification), may improve survival.

We really understand that the use of BCAA as a nutritional supplementation may contribute to the positive effects of LES on survival in patients with cirrhosis. To determine which timing of BCAA supplementation can lead to better survival in patients with cirrhosis, we compared the survival rates between patients treated with LES and those given daytime BCAA supplementation. Among the 523 patients, 157 (30%) received daytime BCAA supplementation and 87 (17%) were given LES supplementation. The results showed that the overall survival rate was significantly higher in the LES group than in the daytime group (median, 9.7 vs. 3.3 years; P < 0.001; Figure 3a), with the HR of 0.40 (95% CI, 0.24–0.63). Of the 231 PS-matched patients, 100 (43%) received daytime BCAA supplementation and 77 (33%) were given LES supplementation. The overall survival rate was significantly higher in the LES group than in the daytime group (median, 5.8 vs. 2.5 years; P < 0.001; Figure 3b). The results showed that LES supplementation significantly improve prognosis, compared to daytime BCAA supplementation (HR, 0.42; 95% CI, 0.24-0.68; P < 0.001; Figure 3b). These results suggest that nocturnal BCAA supplementation, rather than daytime BCAA supplementation, may contribute to the improvement of survival, although the retrospective nature of this study limits the conclusion of the survival benefit of LES therapy. Further prospective studies are therefore required to validate the effect of LES therapy on survival in patients with cirrhosis. To state this point more clearly, we have revised the manuscript, accordingly (lines 221 to 232, 306 to 313; and new Figure 3). Thank you for your good suggestion.

  1. The abstract includes the pre-PS mortality data (rows 19-20); post-PS data would certainly be a better fit.

Based on this suggestion, we have revised the Abstract as follows: Of the 231 propensity-matched patients, 20 (26%) LES-treated patients and 72 (47%) untreated patients died during a median follow-up of 2.0 years (interquartile range, 0.5–4.8) (lines 21 to 23.

  1. The introduction should include a brief definition of LES (no more than a sentence, but expanding beyond de acronym).

In line with the reviewer’s comment, we have provided a brief definition of LES in the Introduction section (lines 50 to 51).

  1. Row 89-90 states that patients were classified as having ascites even though no ascites were detected if they were on diuretic treatment. I find this approach rather peculiar. In light of the current multistate/competing risk approach to advanced liver disease, I am not sure that it adds any clarity.

Based on this comment, we have deleted the suggested sentence. We thank your appropriate suggestion again.

  1. I would suggest expanding the comparative discussion with regards to the study published by Plank L et al. (Hepatology 2008, ref. 17 in the text) since this was a 12-month randomized trial that included more advanced nutritional metrics. 

Plank LD and colleagues have reported a 12-month randomized trial in which they evaluated the effect of LES on nutritional status in patient with cirrhosis. We have added the comparative discussion about the study to the Discussion section. The following description has been added to the manuscript. Plank LD and colleagues have performed a 12-month randomized controlled trial to determine whether nocturnal nutritional supplementation can improve body protein stores in patients with cirrhosis. A total of 103 patients were randomly assigned to daytime or bedtime nutritional supplementation group, and total body protein at baseline, 3, 6, and 12 months were evaluated. Despite the fact that the amount of total body protein at baseline was equivalent in each group, the results have shown that total body protein was significantly increased in the bedtime supplementation group over 12 months, whereas no significant increase in total body protein were noted in the daytime group. Although they showed significant improvement in muscle mass and nutritional status in patients with cirrhosis, they demonstrated no significant benefits in survival, compared to our study. Poor adherence with the prescribed nutritional regimen in their study may explain the poor survival benefit of LES therapy (lines 290 to 300). We thank your valuable comment that improves the quality of our manuscript.

In closing, let me thank you once again for your comments which have helped us to improve the quality of our paper.

Round 2

Reviewer 1 Report

Manuscript improved. My suggestions have been addressed.

Reviewer 3 Report

The revised version of the manuscript significantly improved, as the case for a causal relationship between LES and survival gained strength. Most of my observations were either solved or adequately discussed in the paper. While certainly, a prospective design would be both valuable and interesting, the present form sheds some light and might provide the first steps toward expanding the knowledge in the field.